# 3D Conformational Generative Models for Biological Structures Using Graph Information-Embedded Relative Coordinates

**DOI:** 10.3390/molecules28010321

**Published:** 2022-12-31

**Authors:** Mingyuan Xu, Weifeng Huang, Min Xu, Jinping Lei, Hongming Chen

**Affiliations:** 1Guangzhou Laboratory, Guangzhou International Bio Island, Guangzhou 510005, China; 2School of Pharmaceutical Sciences, Sun Yat-Sen University, Guangzhou 510006, China; 3XtalPi, International Biomedical Innovation Park II 3F, 2, Shenzhen 518000, China

**Keywords:** conformation sampling, generative model

## Abstract

Developing molecular generative models for directly generating 3D conformation has recently become a hot research area. Here, an autoencoder based generative model was proposed for molecular conformation generation. A unique feature of our method is that the graph information embedded relative coordinate (GIE-RC), satisfying translation and rotation invariance, was proposed as a novel way for encoding molecular three-dimensional structure. Compared with commonly used Cartesian coordinate and internal coordinate, GIE-RC is less sensitive on errors when decoding latent variables to 3D coordinates. By using this method, a complex 3D generation task can be turned into a graph node feature generation problem. Examples were shown that the GIE-RC based autoencoder model can be used for both ligand and peptide conformation generation. Additionally, this model was used as an efficient conformation sampling method to augment conformation data needed in the construction of neural network-based force field.

## 1. Introduction

The three-dimensional conformation of a molecule largely influences its biological and physical properties, such as charge distribution, interaction with protein, etc. Furthermore, conformation sampling of complex molecular systems is crucial for calculation of free energy, transition state, and reaction rates.

Traditional conformation sampling methods, including molecular dynamic (MD) simulations [1] and Monte Carlo (MC) [2], are computationally expensive. The sampling efficiency during MD simulation usually suffers from a high free energy barrier, causing repeated sampling near the local minimum and difficulties in converging in high-dimensional space. Many enhanced sampling methods, such as replica exchange molecular dynamics (REMD) [3], accelerated molecular dynamics (AMD) [4], and meta-dynamics simulations [5], can increase the probability of overcoming the free energy barrier and improving sampling efficiency by heating up and modifying the potential energy surface manually. However, complex collective variable settings and high computational costs limit their applications.

Machine learning methods have shed light on sampling problems in high-dimensional space. Various ideas have been proposed to automatically identify the slowest collective variables (CVs) that characterize the structural transformation and to add bias potentials on these CVs for enhanced sampling, including the tICA-RE protocol [6], the combination scheme of genetic algorithms and neural networks [7], the RVAE method [8], etc. On the other hand, deep reinforcement learning was proposed to combine with sampling methods to force the sampling bias towards known space, e.g., reinforced dynamics scheme [9], targeted adversarial learning optimized sampling (TALOS) [10], and reinforced variational adversarial density estimation (RE-VADE) [11]. In addition to the methods mentioned above, deep conformational generative modeling, another active research field in machine learning, has been utilized to compress the high dimensional conformation distribution into low dimensional latent space, thus achieving fast and parallel conformation sampling from learned latent space [12]. Different from a conformation prediction model like Alpha-fold2 [13], the main goal of a conformational generative model is to sample the low-energy structure around different thermostats or interested transition structures [14,15].

Although many important machine learning attempts, including grid-based generation methods [16], distance geometry-based methods [17], and atomic density-based methods [18], were proposed for conformation generation, they could only generate indirect three dimensional information such as a distance matrix, and the final structures were computed by fitting this information or constraining optimization. Recently, Xu et al. proposed the GeoDiff method for conformation generation, in which the core idea is learning to reverse the diffusion process, which recovers the molecule’s geometric distribution from the noisy distribution [15]. Conformation generation based directly on 3D coordinates remains challenging. The difficulty mainly comes from two aspects. First, a molecule can have multiple thermodynamically stable conformations, which makes it difficult to fit a complex multi-modal distribution. Second, there lacks a translational, rotational, and exchange invariant representation for molecular conformation, which is essential for the generalization ability of the model. Recently, the Boltzmann generator [12], a normalized flow-based generative model, was proposed to model a complex BPTI protein’s conformation distribution in the compressed latent space using internal coordinates, where a set of Boltzmann conformations was efficiently sampled in parallel and the free energy difference between two states of BPTI protein could be accurately estimated. Although the Boltzmann generator, as a system-focused conformation generative model, has dramatically overcome the difficulties mentioned above, its versatility is still in question.

Recently, graph-based deep learning models have gained tremendous success in 3D molecular generation due to the fact that they can efficiently generalize the node environment and extract a vector representation of the entire molecule graph. For example, Clevert et al. proposed a graph controlled auto-encoding scheme for molecular 3D representation [19], where internal coordinates were encoded in hidden space independently under the control of the concatenated atomic embedding vectors and succeeded in conformation generation of small ligand datasets by interpolation in the latent space. Unfortunately, the model did not satisfy exchange invariance due to the concatenated operation.

Graph-based autoregressive models have also been proposed for 3D molecular generation, such as L-Net [20], GEN3D [21], cG-SchNet [22], and G-SphereNet [23]. Using these models, the atomic coordinates of a molecule are gradually generated to ensure the rationality of the overall structure, but conformational sampling of the generated compound was not considered. The GEOMOL [24] model, recently proposed by Pattanaik et al., is an end-to-end trainable, non-autoregressive, conformational generative model with a combination of MPNN [25] and self-attention layers [26]. The local structures for each graph node can be predicted and then assembled to yield the molecular conformation, and conformation sampling is performed through multiple predictions on the same molecule graph with random noise, which limits its capability for fitting conformational multi-modal distribution. Generative models for protein design have also been proposed recently, for example, VAE [27] (variational autoencoder)- and GAN [28] (generative adversarial network)-based models were utilized to generate protein 3D structures. Zhang et al. proposed a method of using a relative coordinate system to define the atomic local environment and applied it to predict atomic interactions [29]. Inspired by their work, we proposed a new representation method based on a relative coordinate system to describe the 3D structure of molecules and its application in generative modeling for molecules.

In the current study, a novel translationally and rotationally invariant three-dimensional conformation encoding method, namely graph information-embedded relative coordinate (GIE-RC), was proposed. There are three important attributes of GIE-RC: (1) reversible lossless conversion with 3D coordinates; (2) less sensitive to errors of latent variables compared with other 3D representation methods; (3) convenience in turning a complex 3D generation task into a graph node feature generation problem. A 3D conformational generative model was then constructed by combining the GIE-RC representation with an autoencoder (AE) neural network. Our results show that this model can be used for conformation generation of small molecules as well as large peptide structures. Additionally, this method can be applied as a useful data augmentation tool for improving the construction of a neural network-based force field.

## 2. Results and Discussion

### 2.1. The Advantage of Using GIE-RC for Structure Reconstruction

Under the encoder-decoder generative framework, the decoded structures could deviate from the input conformation due to the reconstruction error of the model. Thus, error resistance in conformation representations could improve the generative model’s reconstruction accuracy. Under the given percentage of noise (error), the conformation reconstruction accuracy with GIE-RC, Cartesian coordinates, and internal coordinates is shown in Table 1. The structure deviation was measured by averaged root mean square deviation (RMSD) on ten pairs of structures. It is worthwhile to mention that the results were based on representation variables directly, no AE model was involved.

Here, lossless conversion to 3D coordinates could be achieved via GIE-RC without any noise. When a given percentage of noise was imposed on the encoded vector, for both coordinates and torsions, the RMSDs between input structures and decoded conformations with the GIE-RC method were smallest among the three structural representation methods (i.e., GIE-RC, internal coordinates ($R_{d,\theta ,\phi }$), and Cartesian coordinates ($R_{x,y,z}$)). Furthermore, the RMSDs of GIE-RC increased modestly as the molecule size increased dramatically, which is especially attractive for generating conformations of large molecules such as peptides and proteins.

### 2.2. Accuracy of Graph-AE for Conformation Reconstruction on Different Datasets

The accuracy of our Graph-AE method was evaluated on the multi-conformation sets of four molecules in Table 1, including two small molecules (PLP and ATP) and two peptides (PDB ID: 2I9M and 1LE1). The structure deviations between Graph-AE reconstructions and the original conformation are shown in Table 2. As a comparison, we also evaluated the accuracy of naive AE reconstruction.

For small molecules, the accuracy of Graph-AE and naive AE was similar, as the corresponding RMSDs between reconstructions and original conformations were 0.39 Å and 0.59 Å for the test set of PLP and 0.9 Å and 0.68 Å for ATP, respectively. However, it was difficult to directly model the conformational distribution of complex protein structures due to high-dimension explosion. Thus, the accuracy of naive AE deteriorated significantly. The coordinate RMSDs for the test sets of 2I9M and 1LE1 were 6.42 Å and 5.05 Å, respectively, and the torsional RMSDs were 46.7° and 40.1° for the test sets of 2I9M and 1LE1, respectively. In these cases, the reconstructed peptide structures collapsed. In contrast, Graph-AE transformed the high-dimensional conformation generation task into a local structure generation task with the constraint of graph embedding, making it easy to model conformation distributions for complex biopolymers. Our results showed that the accuracy of the Graph-AE model was much better than that of the naive AE model.

To further demonstrate the accuracy of the Graph-AE model, the distributions of 12 randomly selected bond lengths and torsional angles of 2I9M are shown in Figure 1 and Appendix A. It can be seen that the 3D coordinates generated by Graph-AE were in good agreement with the ground truth in the test set.

We also evaluated the generalization capability of Graph-AE on the Ligand Depot dataset, as shown in Table 2. The coordinate and torsional RMSDs of reconstructions of unseen ligands in the test set were only 0.78 Å and 10.8°, respectively, which was a decent prediction. Several randomly selected reconstructions that aligned with the original structures are shown in Figure 2. These results demonstrated that the encoded latent space of GIE-RC was generalized enough to decode unseen molecule conformations under the given graph information, enabling conformational reconstructions of unseen molecules. On the other hand, the naive AE model cannot be used to generate conformations for unseen molecules since graph information cannot be used to decode unseen molecules.

### 2.3. Conformation Generation through Latent Space Interpolation

One use of the conformation generative model is to generate diverse conformations for a given compound by sampling directly from the latent space instead of conducting a conformation search. Therefore, we performed latent space interpolation for structure generation and evaluated the structure quality of interpolations. In the case of the PLP dataset, the conformations sampled from a MD trajectory mainly formed two clusters around torsion of $∠D_{C_{4}-C_{6}-O_{3}-P_{1}}$ as shown in Figure 3a,c, which mainly reflected the direction of the phosphoric acid group. The orange cluster represents the main pattern of PLP in MD trajectories, located at $∠D_{C_{4}-C_{6}-O_{3}-P_{1}\;}of$ 180°. The gray cluster, located at $∠D_{C_{4}-C_{6}-O_{3}-P_{1}}\;of$ 60°, represents a local energy minimum of PLP. We performed DBSCAN clustering to remove the noisy structures scattered between the two clusters of PLP. Then, we retrained both the Graph-AE and naive AE models with the conformations of these two clusters, where the transition area between the clusters was not included in the training set. Twenty pairs of start and endpoints were randomly selected to perform linear interpolation in the latent space. The blue dots in Figure 3a,c represent the interpolated conformations. The representative interpolations and distribution of randomly selected internal coordinates of PLP are shown in Figure 4 and Appendix A. It can be seen that the torsion of $∠D_{C_{4}-C_{6}-O_{3}-P_{1}}$ was continuously changing among the interpolations of the Graph-AE- and CNN-based AE models, indicating the continuity of the GIE-RC latent space. Interestingly, the pyridine group remained unchanged among the interpolated conformations, meaning there was clear correspondence between the latent space and physical meaningful internal coordinates.

The ATP case was more complicated, in which the initial conformations were clustered around two torsions of $∠D_{P_{A}-O_{5^{\prime }}-C_{5^{\prime }}-C_{4^{\prime }}}$ and $∠D_{O_{5^{\prime }}-C_{5^{\prime }}-C_{4^{\prime }}-O_{4^{\prime }}}$ as shown in Figure 3b,d. The left-top cluster represents a metastable state of ATP, and the bottom-right cluster represents the main cluster of conformations in the MD simulations. Similarly, DBSCAN-based clustering and model retraining were also performed to ensure that the transition area between two clusters was not included in the training set. As shown in Figure 4b, the representative interpolations represented the transitional conformations of ATP, where the ribose ring and triphosphoric acid group transformed from a ‘S’ shape to a ‘C’ shape.

Similar procedures were carried out for the 1LE1 and 2I9M datasets. Here, clusters having more than 1000 conformations were kept to represent the primary clusters of the MD simulation and the other minor conformations were removed from the datasets. After retraining the Graph-AE models for 1LE1 and 2I9M, pairwise interpolations between all cluster pairs were performed to evaluate the latent spaces which were not included in the training set. Similarly, interpolation was carried out among 20 random pairs of start and endpoints. As shown in Figure 5, the kernel density estimation (KDE) distribution of the remained 1LE1 and 2I9M conformations and the interpolated conformations were mapped on the 2D plane using the multi-dimension scaling (MDS) method [30]. It can be seen that interpolations filled the empty area between the clusters. For 1LE1, interpolation was carried out among all six major clusters. This demonstrated that the GIE-RC latent spaces were continuous for given peptides. Some representative interpolated conformations between the first and second clusters are shown in Figure 6, which represented a gradual folding process from a twisted linear peptide into a deformed β-sheet for 1LE1 and an α-helix conformation for 2I9M, respectively. The distribution of randomly selected interpolations was in good agreement with that of the conformations of MD simulations (as shown in Appendix A). These results demonstrated that the GIE-RC-based Graph-AE model could be a useful tool for sampling meaningful conformations in transitional states.

### 2.4. Structure Augmentation for Neural Network-Based Force Field Constructions

In recent years, machine learning-based force field, marrying the accuracy of QM methods with the efficiency of MM force field, has been a hot research area in MD simulation and material design, and numerous models have been proposed [29,31,32]. Among these efforts, the construction of reference dataset is crucial for accuracy of the model, as the dataset should cover the targeted chemical space as much as possible and its size should be as small as possible. Normally, an MD sampling operation is necessary for data augmentation and an active learning scheme is also used for removing redundancy in the dataset [33,34,35].

The function of the GIE-RC-based Graph-AE model in fast conformation sampling makes it a useful data augmentation tool for constructing a neural network-based force field. As an example, we applied the Graph-AE model to quickly build an ANI-1 type neural network potential energy surface (NNPES) for the Ace-ALA-Nme peptide using the default setting of the Torch-ANI package [36]. First, 6000 conformations were extracted from a 60 ns MD simulation trajectory in an explicit water model and then used for training an ANI-1 neural network potential at the PM6 level. The standard PM6 potential energy surface (PES) and initial ANI-1 neural network potential energy surface (NNPES) among the dihedral of $∠D_{C_{0}-N_{1}-C_{\alpha ,1}-C_{1}}$(ϕ) and $∠D_{N_{1}-C_{\alpha ,1}-C_{1}-O_{1}}$ (ψ) are shown in Figure 7a,b. The RMSE (root mean square error) between initial ANI-1 NNPES and PM6 PES was 0.065 eV and the location of the initial ANI-1 NNPES global minima was incorrect. These deviations were mainly due to insufficient sampling in the transition region. The ANI-1 NNPES could be gradually improved with the extension of MD sampling and finally converged by running a 200 ns MD simulation and sampling with 20,000 structures, as shown in Figure 7c. The details for building NNPES are discussed in Section 3.3, here we mainly focus on how to augment our dataset of NNPES training.

Instead of enhancing the sampling by running a computationally expensive MD simulation to improve the quality of the ANI-1 NNPES, we chose to train the Graph-AE model with the initial 6000 MD sampled conformations of the Ace-ALA-Nme peptide, and then performed pairwise interpolations between 4 major clusters of the 6000-conformation set to augment our dataset for building NNPES. The interpolation was performed among 300 randomly selected pairs of start and endpoints with 20 equal interval interpolation steps. Interpolated structures with energy values within 27 eV (1 Hartree) to the lowest value of the initial dataset were added to the NNPES dataset and the ANI-1 potential energy surface was then updated.

In total, the final dataset was augmented to 17,898 structures and the RMSE between updated ANI-1 NNPES and PM6 PES was 0.044 eV, which was close to the accuracy of ANI-1 NNPES built on conformations sampled from a 200 ns MD (RMSE 0.035 eV). Our result clearly showed that the accuracy of the updated ANI-1 NNPES enhanced by the conformations sampled from the Graph-AE model was significantly improved without running another extended MD sampling. The 2D PES landscape is shown in Figure 7d. The global minima of the Graph-AE model updated ANI-1 NNPES was around (−100°, 50°) and (75°, −60°), which was in a good agreement with those of standard PM6 PES. It can also be seen that the local landscape of the high energy region was quite similar to the standard PM6 PES. The local minima around (75°, 50°) were also well predicted in the AE updated ANI-1 NNPES. As a comparison, we also computed the PES of our toy system under Amber FF14SB and MMFF94 force fields, as shown in Figure 7e,f. In Amber FF14SB PES, the global minima were around (0°, 0°), which is the global maximum in PM6 PES. However, it is surrounded by high barriers, which makes the system hardly enter this region in MD simulations. Local minima around (−100°, 50°), (75°, −60°), and (75°, 50°) also exist. However, the potential energy in the $∠D_{C_{0}-N_{1}-C_{\alpha ,1}-C_{1}}>100{}^{\circ}$ region was significantly larger than its in $∠D_{C_{0}-N_{1}-C_{\alpha ,1}-C_{1}}<-100{}^{\circ}$, which disagreed with PM6 PES. The shape of the MMFF94 force field is similar to that of PM6 PES. However, it also predicted the global minima around (0°, 0°), similar to Amber FF14SB. ANI-1 NNPES provided a better approximation to PM6 PES. In summary, these results suggested a potential application of Graph-AE as a data augmentation tool of NNPES construction.

## 3. Methods and Materials

### 3.1. Generating Graph Information-Embedded Relative Coordinate (GIE-RC) for Molecular Conformation

A given molecule’s 3D structure is usually expressed as a matrix composed of atom types and coordinates. Its chemical properties should keep constant when its matrix representation is changed due to atom order exchange or coordinate translation or rotation. The derived machine learning model should be independent from these operations, which is also called exchange invariance and SE (3) invariance. Various 3D representations were previously proposed. For example, Xu [17] used a distance matrix to represent 3D information, an array of fragment coordinates together with their torsional information was used in GeoMol [24] for conformation representation, and explicit 3D coordinate representation was used in GeoDiff [15] model. For most of these reported generative models, reconstruction of coordinates was based on a global coordinate system, some of which did not necessarily satisfy exchange invariance and SE (3) invariance. In contrast, we propose a graph information-embedded relative coordinate model (GIE-RC) for representing molecular conformation, in which molecular 3D coordinates are encoded into atomic environment features by constructing a local coordinate system and derived information from a 2D topology graph. In this way, exchange invariance and SE (3) invariance is guaranteed. On the other hand, this feature can be lossless converted to molecular 3D coordinates.

In the GIE-RC framework, assuming a local coordinate system is set up on a specific atom, its atomic environment feature can then be defined by a 2D matrix composed by the relative coordinates (RC) of neighboring atoms in the local coordinate system (i.e., the local structure feature). For each molecule, such an atomic environment feature satisfies exchange, translational, and rotational invariance. The GIE-RC encoding workflow of a molecule with *N* atoms contain five steps, as shown in Figure 8.
(1)The binary 2D adjacent matrix of a molecule is converted to a weighted graph distance matrix $R^{2D}$ by first calculating the shortest bond distance between the atom pair using the Dijkstra algorithm [37], then summing the standard bond length in the MMFF94 force field [38] as the distance of the atom pair in the graph. In this way, the element in the $R^{2D}$ matrix is the weighted bond distance between the atom pair.(2)A Coulomb matrix [39] M is constructed based on the weighted graph distance matrix $R^{2D}$, the element of the matrix $M_{ij}$, as defined in Equation (1):
(1)$$M_{ij}=\left\{\begin{array}{ll} 0.5Z_{i}^{2.4} & \forall i=j\;\\ \frac{Z_{i}Z_{j}}{r_{ij}^{2D}} & \forall i\neq j \end{array}\right.\;\;\;\;$$
where $Z_{i}$ is the atomic number of atoms i and $r_{ij}^{2D}$ is the 2D weighted graph distance between atom i and j. Then, a unique order S of atoms is determined by sorting the eigen values [$\lambda_{1},\lambda_{2}\ldots \lambda_{n}$] of Coulomb matrix *M*. (3)Calculation of the relative coordinates of neighbor atoms within three bond distances of center atom i. Here, atoms *a* and b (as shown in Figure 9) are the closest atoms to atom *i* in the weighted 2D graph. If there are atoms which have the same distance to atom *i*, their order is decided based on the order S. $C_{j}^{i}$ is the relative coordinate of atom *j* in the local coordinate system determined by i,a,b and defined in Equation (2):
(2)$$C_{j}^{i}=\left[\frac{S_{2D}\left(r_{ij}^{2D}\right)X_{j}^{i}}{r_{ij}^{2D}},\frac{S_{2D}\left(r_{ij}^{2D}\right)Y_{j}^{i}}{r_{ij}^{2D}},\frac{S_{2D}\left(r_{ij}^{2D}\right)Z_{j}^{i}}{r_{ij}^{2D}}\right]$$
where {$X_{j}^{i},Y_{j}^{i},Z_{j}^{i}$} are the local 3D coordinates of atom j in the atom i centered coordinate system. $r_{ij}^{2D}$ is the weighted 2D graph distance between atom i and *j*. $S_{2D}\left(r_{ij}^{2D}\right)$ is an attenuation factor defined in Equation (3), ensuring that the contribution of distant atoms to the relative coordinates is gradually reduced to 0. $r_{cs}$ and $r_{c}$ are user defined cut-off value set to 5.0 and 6.0, respectively, in the current study.
(3)$$S_{2D}(r_{ij}^{2D})=\{\begin{array}{cc} \frac{1}{r_{ij}^{2D}} & r_{ij}^{2D}<r_{cs}\\ \frac{1}{r_{ij}^{2D}}*\left(0.5*\cos\left(\frac{\pi \left(r_{ij}^{2D}-r_{cs}\right)}{r_{c}-r_{cs}}\right)+0.5\right) & r_{cs}<r_{ij}^{2D}<r_{c}\\ 0 & r_{ij}^{2D}>r_{c} \end{array}$$(4)Aggregate the RCs of neighboring atoms around central atom i to obtain the local structure feature $F_{i}$ (ie. GIE-RC) as the environment feature for atom i, which represents the local environment of atom i. The RCs of local structure feature $F_{i}$ are sorted by the weighted graph distance to atom i in increasing order.
(4)$$F_{i}=\left[C_{1}^{i},C_{2}^{i},\ldots .C_{m_{i}}^{i}\right]$$

Here, $m_{i}$ is the number of neighbor atoms around atom i within three bond distances. When the calculation of $F_{i}$ for all atoms is done, the $F_{i}$ tensor will be padded to the same dimension of *m* with zeros, where *m* is the maximal neighbor atoms in the local structure among all molecules of the dataset. Additionally, the order of two atoms with the same graph distance to center *i* is determined by the standard order ***S*** of the Coulomb matrix M in GIE-RC. Since the $r_{ij}^{2D}$ and $S_{2D}\left(r_{ij}^{2D}\right)$ terms in Equation (3) are determined from the 2D graph, the internal coordinates within the local structure around each central atom i can be reversibly restored from the atomic environment feature $F_{i}$. In this way, we get a reversible SE (3) invariant description of the local structure around any atom in the graph.

(5)All atomic environment features ($F_{i}$) are sorted by the standard order S and compose a unique feature tensor ***F*** to represent the conformation, named as the graph information-embedded relative coordinate matrix (GIE-RCM). After the encoding process, every molecular conformation can be represented by a set of atomic environment features {$F_{i}$}, i.e., $F_{i}$ refers to the local structure feature of atom i.

As shown in Figure 1b, assuming that a molecular conformation is composed of a set of internal coordinates $\left[ic_{1},ic_{2},ic_{3}\ldots ic_{3N}\right]$ (N is the number of atoms), the final internal coordinates of the molecular conformation are then obtained by calculating the average internal coordinates {$ic_{i}$} over all the local structures:(5)$$ic_{i}=1/k\sum_{j=1}^{k}ic_{i}^{j}$$
assuming $ic_{i}$ existed in k local structures and $ic_{i}^{j}$ refers to the obtained $ic_{i}$ in the atom *j* centered local structure.

### 3.2. Graph Constrained Autoencoder Model for Structure Generation

As mentioned above, the GIE-RC framework can turn the 3D conformation generation task into a graph node feature generation. A graph-constrained autoencoder (Graph-AE) model was utilized for conformation generation, as shown in Figure 10.

The Graph-AE model contains two encoder modules, one is the graph convolution network (GCN) to generate 2D graph information around each atom, and the other one is the encoder for the GIE-RC-based local structure feature $F_{v}$, which contains 3D information of the neighboring atoms around each atom, as described in the previous section. In general, a given molecule can be expressed as a graph $G=\left(V,E\right)$ including the features for all nodes $x_{v}$, ∀v∈V and features for all edges $x_{e_{v,w}},\;\;\forall e_{v,w}\in E$. For a given node *v*, its initial node feature $x_{v}$ is concatenated from the one-hot encoding of its atomic number, atom degree, formal charge, chirality, and hybridization state. The edge feature $x_{e_{v,w}}$ is defined by the one-hot encoding of the bond type between node v and w. The initial embedding of $x_{v}\;$ and $x_{e_{v,w}}$ in a graph are annotated as $h_{v}^{0}$ and $h_{e_{v,w}}^{0}$. After passing through a modified message passing neural network, CMPNN [40], which can improve the molecular graph embedding by strengthening the message interactions between edges and nodes, the node embeddings $h_{v}$ are extracted as the representation of graph information around node *v*. To be consistent with the definition of local structure feature $F_{v}$, three layers of CMPNN were set to generate graph information in three bonds distances of each atom.

3D coordinates of the input molecule were encoded to RC-based local structure features $\left\{F_{v}\right\}$ (as illustrated in Figure 8a). A simple multi-layer MLP module was then used to transform $F_{v}$ to latent vector $z_{v}$. To improve the generalization of the AE model, gaussian noise $N_{g}$ was added to $z_{v}$. The graph embedded $h_{v}$ and the GIE-RC-related embedded $z_{v}$ were concatenated and fed into the decoder module. The decoder tried to reconstruct $F_{v}$ for each node. The whole encoding-decoding process can be expressed as Equations (6)–(8).
(6)$$h_{v}=CMPNN\left(v,G\right)$$
(7)$$z_{v}=Encoder\left(F_{v}\right)$$
(8)$$F_{v}^{o}=Decoder\left(CONCAT\left(h_{v},z_{v}+N_{g}\right)\right)$$

In Graph-AE, the loss function was calculated based on mean standard error (MSE) as following equation:(9)$$L_{Graph-AE}=MSE\left(F_{v}^{o},F_{v}\right)$$

As a comparison, a naive AE model (without graph constraint) was also trained to directly reconstruct the matrix F of GIE-RC. The encoder contained a CNN module for GIE-RC feature extraction and an MLP for encoding latent vectors. Here, the decoder network was symmetrical to the encoder, and the model was also optimized with the MSE loss between $F_{v}^{o}$ and $F_{v}$.

### 3.3. Datasets and Computational Settings

As shown in Figure 3b, the Graph-AE model contained a GCN module including three message-passing layers. Its encoder and decoder modules had the same architecture, and each contained three dense layers with 256 neurons. In the CNN-based naive AE model, the encoder comprised three 2D convolutional layers and two dense layers with 128 neurons. The decoder contained two dense layers with 128 neurons and five 2D transposed convolutional layers for better reconstruction.

Both models used the Tanh activation function and were optimized with the Adam algorithm and decayed learning rate. The dimension of latent space was 128. The training process stopped when learning rates dropped to 10^−8^. We adopted an early stop strategy to save computational costs. The percentage noise of the Graph-AE and CNN-based AE models was set to 5% to improve the generalization of the model. All training was performed on single iGame GeForce RTX-2080ti GPU (Shenzhen, China).

For calculating GIE-RCs, the user-defined cutoffs $R_{cs}$ and $R_{c}$ in Equation (4) were set to 5.0 Å and 6.0 Å, respectively. In the decoding stage, the DBSCAN clustering algorithm [41] in Scikit-learn [42] was used to remove noise or unreasonable values. In each dimension, only the biggest cluster was used to obtain average internal coordinates among different local structures around each atom. The eps parameter of DBSCAN for bond distance, angle, and dihedral were 0.1 Å, 20°, 20°, respectively.

We evaluated the conformation reconstruction performance of the Graph-AE and naive AE models on four conformation datasets, including drug-like molecules and peptides (as shown in Table 1). The PLP and ATP datasets were compounds with 2500 and 5000 conformations, respectively. They were separately extracted from 50 ns and 100 ns MD trajectories in TIP3P explicit water models at equal intervals. The 2I9M and 1LE1 datasets were two peptides with α-helix and β-sheet conformations, respectively. The x-ray structures were obtained from the Protein Data Bank (PDB) database (http://www.rcsb.org/pdb/, accessed on 20 December 2021). Each dataset contained 20,000 conformations, extracted from 500 ns folding MD trajectories in explicit water. Additionally, the Ligand Depot dataset [43], including 36,244 ligands from the PDB Bank [44], was used to evaluate the generalization capability of Graph-AE. For all datasets, the training set and test set were obtained by randomly spliting at a ratio of 9:1.

All MD simulations carried out in the current study (including PLP, ATP, 2I9M, 1LE1, and the Ace-ALA-Nme peptide) were performed with the Amber 18 package [45]. The solute was solvated in a TIP3P water box with a radius of 20 Å. The peptide and ligand force field parameters were from the FF14SB and GAFF force fields, respectively, and Langevin dynamics drove simulations in the NPT ensemble with a collision frequency of 2 ps^−1^. Additionally, the QM calculations of the Ace-ALA-Nme peptide were carried out with the Gaussian 16 [46] package at the PM6 level.

The ANI neural network potential energy surface (NNPES) model was trained using the default setting of the Torch-ANI package [36]. Each ANI NNPES model contained 4 subnets for elements of H, C, N, and O, respectively. The subnet consisted of 3 hidden layers with 500 neurons. Both the energy and force information were used in the training of NNPES with RMSE loss between predictions and references. AdamW [47] and SGD [48] algorithms were used to optimize the weight and bias of subnets, respectively. The learning rate decay in the training process was determined with the ReduceRLOnPlateau algorithm and the training process stopped when the learning rate was less than 10^−6^. As a comparison, MMFF94 [38] and Amber FF14SB [49] force field PES were also provided, which were computed with RDKit [50] and OpenMM [51], respectively.

## 4. Conclusions

Here, we propose a novel structure representation method for a 3D generative model, named GIE-RC, which is insensitive to errors and satisfies translational and rotational invariance. It was used to encode molecular 3D coordinates in a reversible and lossless manner. By combining the GIE-RC concept with the graph-constrained autoencoder model, a complex 3D generation task can be turned into a node feature generation problem. Several examples were shown, demonstrating that the GIE-RC-based Graph-AE model can be used to efficiently sample molecular conformations of small molecules, as well as peptides, through interpolation in the latent space. We also applied the Graph-AE model for sampling peptide conformations to build an ANI-1 neural network potential energy surface, and our results showed that the derived NNPES using the conformations sampled from the Graph-AE model were the same quality as the model with expensive MD simulations.

## Figures and Tables

**Figure 1 molecules-28-00321-f001:**
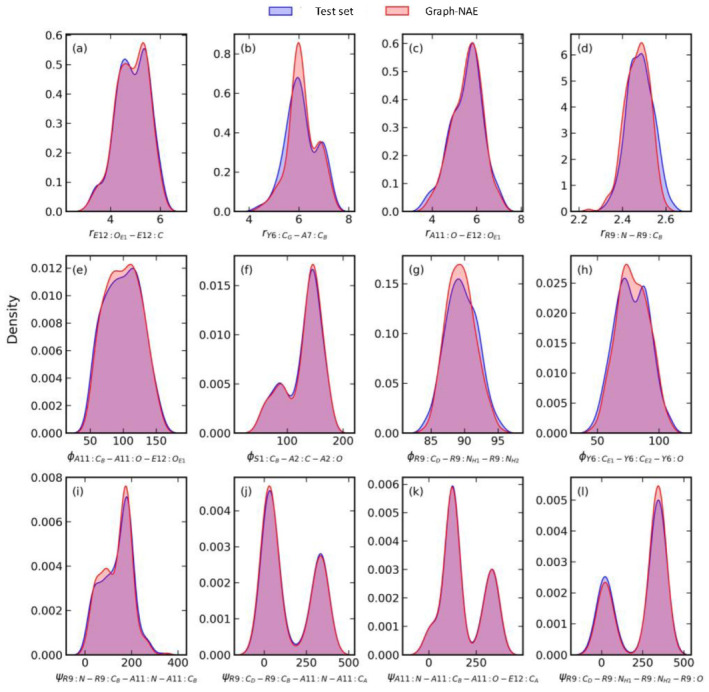
The distribution of 12 randomly selected pair distances, bond angles, and torsional angles of 2I9M (**a**–**l**) in Graph-AE reconstructed conformations (**red**) and their ground truths in the test set (**blue**).

**Figure 2 molecules-28-00321-f002:**
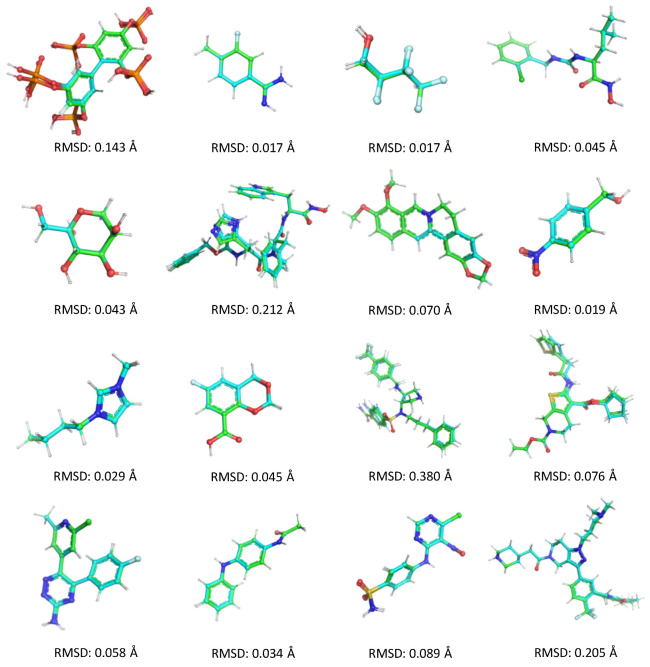
Aligned structures of reconstructions and original conformation of unknown molecules in Ligand Bank dataset.

**Figure 3 molecules-28-00321-f003:**
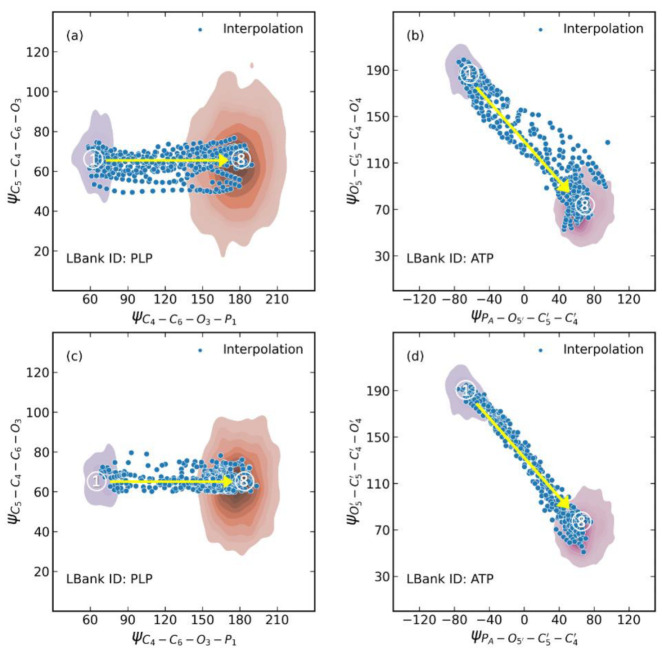
Distribution of interpolated conformations in the latent space of (**a**) PLP naive AE model; (**b**) ATP naive AE model; (**c**) PLP Graph-AE model; (**d**) ATP Graph-AE model.

**Figure 4 molecules-28-00321-f004:**
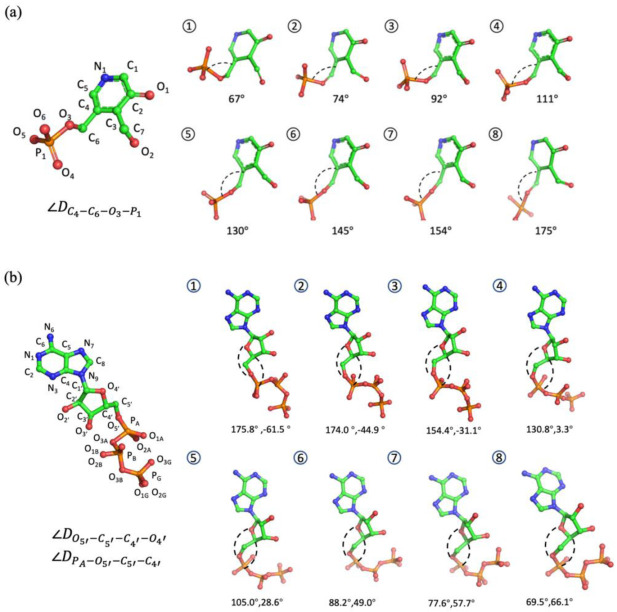
Eight representatives of the interpolated structures (①–⑧) of (**a**) PLP and (**b**) ATP.

**Figure 5 molecules-28-00321-f005:**
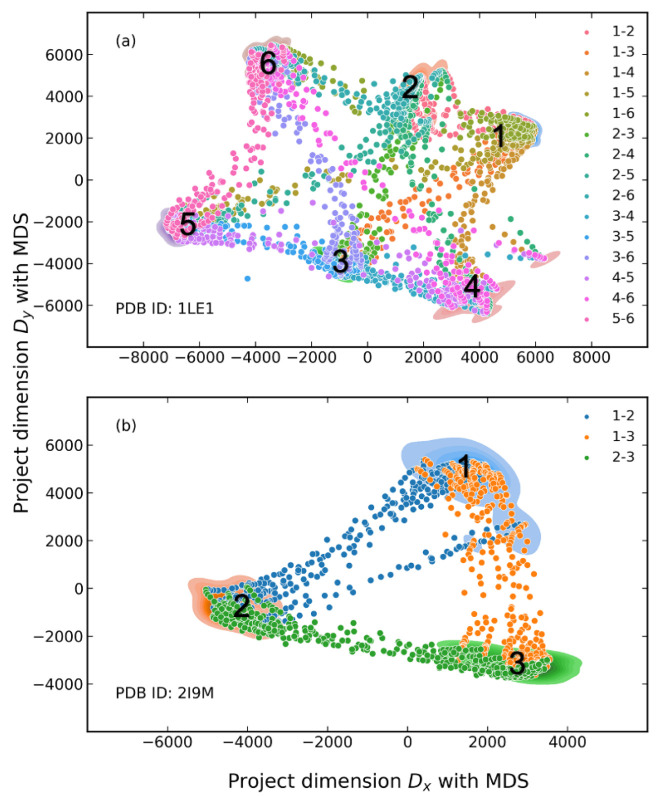
Distribution of primary patterns of datasets and conformations sampled via latent space interpolations. (**a**) Cross interpolations between six clusters of 1LE1 with Graph-NAE, (**b**) cross interpolations between three clusters of 2I9M with Graph-NAE.

**Figure 6 molecules-28-00321-f006:**
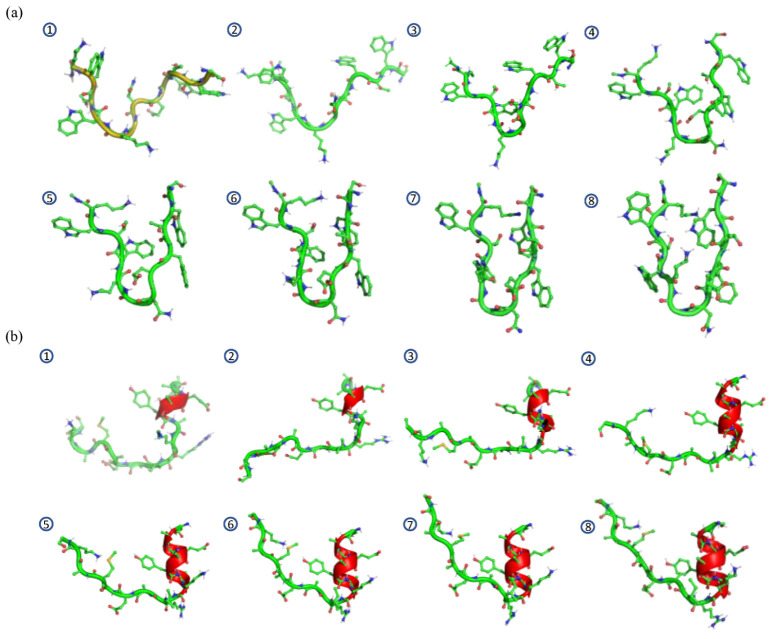
Eight representative interpolation structures between cluster 1 to cluster 2 (①–⑧) of (**a**) 1LE1 and (**b**) 2I9M.

**Figure 7 molecules-28-00321-f007:**
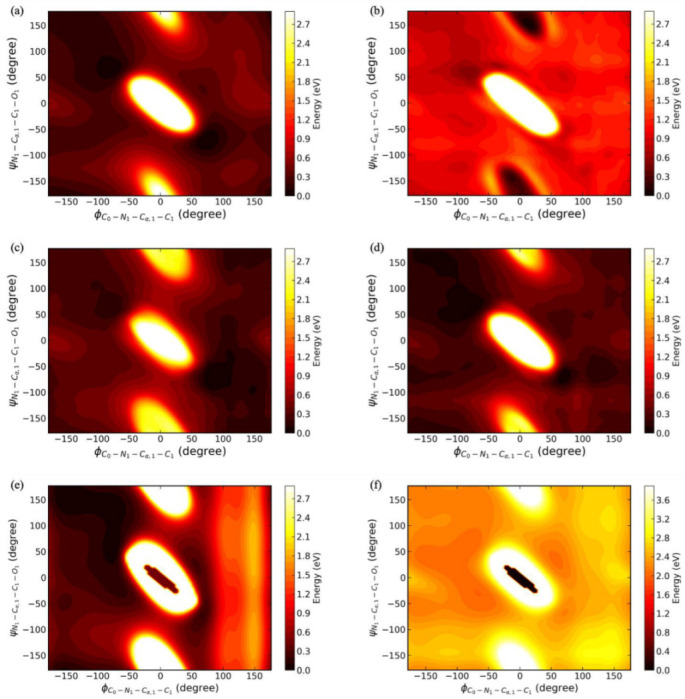
Potential energy surface (PES) of the Ace-Ala-Nme peptide among the dihedral of $∠D_{C_{0}-N_{1}-C_{\alpha ,1}-C_{1}}$
(ϕ) and $∠D_{N_{1}-C_{\alpha ,1}-C_{1}-O_{1}}$ (ψ) (**a**) Standard PES of PM6, (**b**) initial ANI-1 NNPES trained at PM6 level with 60 ns MD sampling, (**c**) converged ANI-1 NNPES with 200 ns MD sampling, (**d**) ANI-1 NNPES updated with interpolation structures of Graph-NAE, (**e**) PES of Amber FF14SB force field, (**f**) PES of MMFF94 force field.

**Figure 8 molecules-28-00321-f008:**
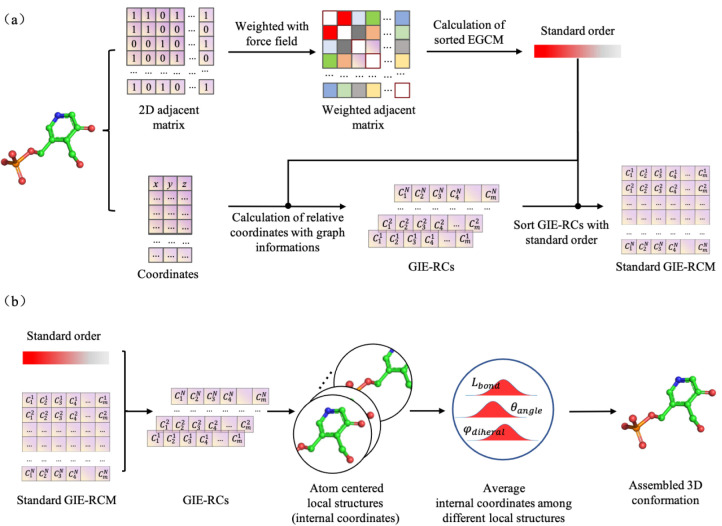
Structure encoding (**a**) and decoding (**b**) via GIE-RC.

**Figure 9 molecules-28-00321-f009:**
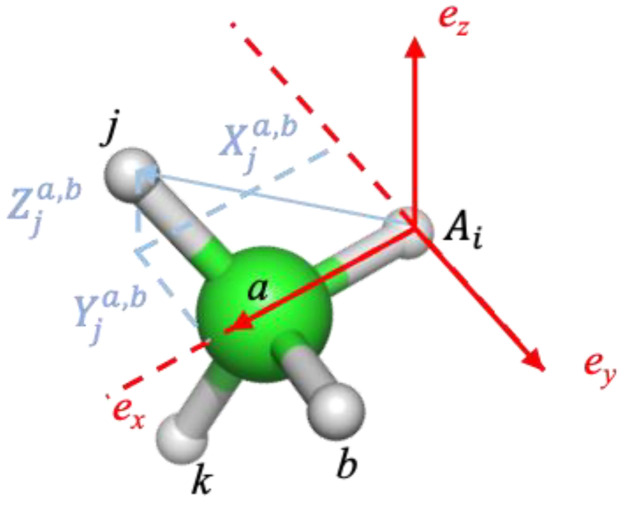
Definition of relative coordinates of atom *i*’s graph node neighbors including *a*, *b*, *j*, *k*.

**Figure 10 molecules-28-00321-f010:**
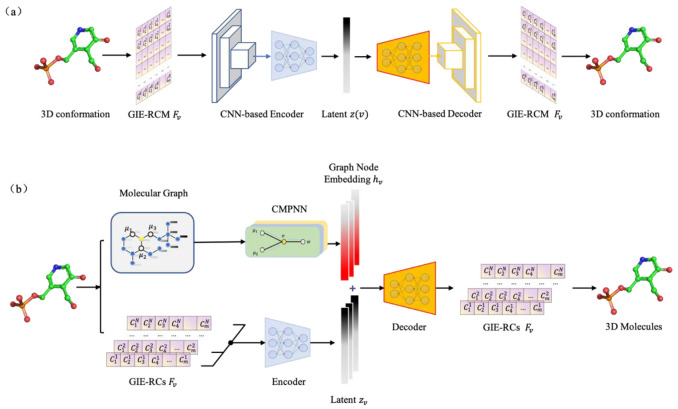
Detailed structure of (**a**) CNN-based naive AE model and (**b**) Graph-AE model.

**Table 1 molecules-28-00321-t001:** Average structure deviation between input structures and reconstructed structures based on the representation variables given different percentages of noise.

System	Structures	No of Atoms	% of Noise	RMSD (Å)	Torsion RMSD (°)
GIE-RC	Rd,θ,ϕ	Rx,y,z	GIE-RC	Rd,θ,ϕ	Rx,y,z
PLP	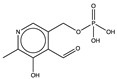	24	0.0	0.00	0.00	0.00	0.00	0.00	0.00
2.5	0.11	0.43	0.63	0.96	4.13	48.91
5.0	0.19	1.11	1.15	2.09	9.55	71.96
10.0	0.47	2.14	2.31	3.67	16.72	84.03
ATP	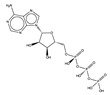	43	0.0	0.00	0.00	0.00	0.00	0.00	0.00
2.5	0.21	0.87	1.09	0.93	4.78	78.98
5.0	0.48	2.41	2.14	2.1	8.92	94.07
10.0	1.20	4.27	4.23	4.49	18.43	99.13
1LE1	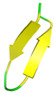	145	0.0	0.00	0.00	0.00	0.00	0.00	0.00
2.5	0.89	5.08	1.47	1.08	5.51	81.71
5.0	1.45	7.18	2.94	2.16	10.48	98.3
10.0	2.92	14.84	5.86	3.76	21.57	105.49
2I9M	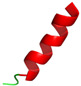	150	0.0	0.00	0.00	0.00	0.00	0.00	0.00
2.5	1.12	6.43	1.79	1.19	5.18	87.48
5.0	2.73	13.56	3.42	2.34	10.79	102.92
10.0	5.20	22.06	6.97	4.51	21.14	107.92

**Table 2 molecules-28-00321-t002:** The average RMSD between original structures and reconstructions among different models on PLP, ATP, 2I9M,1LE1 and Ligand Depot datasets.

System	Dataset	Graph-AE		Naive AE	
		RMSD (Å)	Torsion RMSD (°)	RMSD (Å)	Torsion RMSD (°)
PLP	Training set	0.22	6.68	0.22	5.73
	Test set	0.39	15.80	0.59	18.46
ATP	Training set	0.59	10.46	0.30	10.74
	Test set	0.9	17.69	0.68	15.23
2I9M	Training set	0.85	2.35	4.25	38.59
	Test set	0.91	2.40	6.42	46.74
1LE1	Training set	0.74	2.30	3.80	38.83
	Test set	0.76	2.31	5.05	40.07
Ligand Depot	Training set	0.69	9.46	/	/
	Test set	0.78	10.18	/	/

## Data Availability

Not applicable.

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
