# Peer review of "3D Conformational Generative Models for Biological Structures Using Graph Information-Embedded Relative Coordinates"

_molecules, 2022, doi:10.3390/molecules28010321_

Round 1

Reviewer 1 Report

In this paper, the authors proposed machine learning- based method for molecular conformation generation, and then the application showed that this method can be used for both ligand and peptide conformation generation. This is a quite interesting work, and the proposed method and results are also quite good. This paper is well written, and it can be accepted by molecules by including the following minor comment.

The idea or concept of neural network or machine learning -based force field is relatively new for me. But it would be interesting for physical chemist, so I would like to more direct comparison or discussion of this between physical-based force field. When I check the references 33-38, it still confused for me.

Reviewer 2 Report

The manuscript entilted "3D conformational generative models for biological structures using graph information embedded relative coordinates" by Mingyuan Xu, Weifeng Huang, Min Xu, Jinping Lei, and Hongming Chen, presents an autoencoder based generative model (Graph-AE model) for molecular conformation generation.

An original feature of this encoder is to be based on graph information embedded relative coordinates (GIE-RC) which are derived by constructing local coordinate system from 2D topology graph.

This approach is applied to the analysis of various molecular objects: (a) the conformations of two ligands generated by 50ns and 100ns MD trajectories in explicit water; (b) the conformation of an alpha and a beta peptide generated by 500 ms folding trajectories;

The Graph-AE model allows to reconstruct the conformations of these molecules, and the graph feature is essential for improving the reconstruction.

In addition, a test of Graph-AE on the Ligand Depot dataset proves the ability to generalization of the model. In addition, Graph-AE is able to interpolate the conformational transitions of the ligands and the peptides. Finally, the ability of Graph-AE to reconstruct the potential energy surface (PES) of Ace-Ala-Nme was demonstrated.

The manuscript is interesting, but it would gained a lot by improving the precision of the description. Indeed, several expression are not defined, as: 3-hop distance,

Torsion RMSD (why the torsion RMSD is so large in Table 1?). Also, it would be interesting to have a bit more information about the generation of NNPES (the reference 33 seem not to be connected to NNPES).

In the Equation 3, it seems that r_cs is smaller than r_c, but r_cs=6.0 and r_c)= 5.0 Angstroems.

At line173, the sentence: "Obviously, by doing in this way, the 3D information of neighbor atoms around atom i is defined in the local structure feature F" is not clear.

At line 420, in the expression: "can be referred to literature 43", 43 should be put in brackets.

The manuscript should undergo a major revision in order to make it more clear.
